# The Impact of COVID-19 on Hyperhidrosis Patients in the Mental Health and Quality of Life: A Web-Based Surveillance Study

**DOI:** 10.3390/jcm11133576

**Published:** 2022-06-21

**Authors:** Wongi Woo, Jooyoung Oh, Bong Jun Kim, Jongeun Won, Duk Hwan Moon, Sungsoo Lee

**Affiliations:** 1Department of Thoracic and Cardiovascular Surgery, Gangnam Severance Hospital, Yonsei University College of Medicine, Seoul 06273, Korea; woopendo@gmail.com (W.W.); claris@yuhs.ac (B.J.K.); whddms94@yuhs.ac (J.W.); pupupuck@yuhs.ac (D.H.M.); 2Department of Psychiatry, Gangnam Severance Hospital, Yonsei University College of Medicine, Seoul 06273, Korea; ojuojuoju@yuhs.ac

**Keywords:** hyperhidrosis, COVID-19, depression, PHQ-9, online survey

## Abstract

Background: We aimed to investigate the impact of the COVID-19 pandemic on the degree of depression among hyperhidrosis patients and their quality of life. Methods: 222 patients were contacted through an online questionnaire. Patients reported quality of life (QoL), including treatment and changes in symptoms during the pandemic, and also responded to the Patient Health Questionnaire-9 (PHQ-9) to evaluate the severity of depression. Those were compared with the result from the general population. Spearman correlation and multiple linear regression were performed to identify the factors related to the PHQ-9 score. Results: Half of the patients were female. The mean PHQ-9 score (5.25) of hyperhidrosis patients was higher than the general population, and female patients displayed significantly higher PHQ-9 scores than males (*p* = 0.002). QoL was impaired more in females. About 10% of patients experienced worsening symptoms, and 30% had difficulties getting appropriate management. Significant negative correlations were found between the PHQ-9 and age or disease duration. Predictive factors for the PHQ-9 were female (*p* = 0.006) and facial hyperhidrosis (*p* = 0.024). Conclusions: The level of depression among hyperhidrosis patients was higher than the general population during the COVID-19 pandemic; female and facial hyperhidrosis patients need much more psychiatric attention. Though hyperhidrosis is classified as benign and often neglected by clinicians, we need to give more awareness to the mental burden imposed by the COVID-19 pandemic.

## 1. Introduction

The COVID-19 pandemic has drastically changed the daily lives of people around the world, causing mental health problems in many. Evidence from previous research suggests that the prevalence of depression has increased during the pandemic [1,2,3,4,5]. Among health care workers, the rate was as high as 50.4% in China [1] and 63.5% in India [6]. Other than these frontline workers, the general public also saw an increase in the prevalence of depression: 17.5% in South Korea [3] and 27.8% in the United States [2]. Due to the increased burden of mental distress, the pandemic may have had a greater impact on mentally vulnerable patients [7].

Mental health problems, such as depression and anxiety, in hyperhidrosis patients, have been investigated in previous studies [8,9,10,11,12]. Although patients with hyperhidrosis lack overt psychopathology, their complaints about mental distress appear to be associated with the degree of symptoms and quality of life [13]. Treatments, such as endoscopic sympathetic surgery [14,15,16] and botulinum toxin injection [17], have shown to significantly improve the symptoms and mental health-related problems. However, the level of evidence is still insufficient due to the limited number of studies; additionally, a recent report on dermatologic disease revealed that patients with dermatologic conditions faced barriers to having appropriate management during the pandemic [18]. 

In this study, we aimed to investigate the level of depression among hyperhidrosis patients during the pandemic and explore possible risk factors related to a higher level of depression. Additionally, this research investigated how the COVID-19 pandemic impacted the quality of life and management in hyperhidrosis patients.

## 2. Methods

### 2.1. Study Population and Design 

Participants were contacted from the largest online community (Community of people overcoming excessive sweating, https://m.cafe.daum.net/hyperhydrosis, accessed on 15 July 2021) of hyperhidrosis patients in South Korea, which has 36,015 registered members and 120 average daily visits. The online questionnaire with informed consent was distributed to all members of this community.

### 2.2. Online Survey 

The online survey was performed using Google Forms, and code numbers were used to avoid disclosing patients’ names or other personally identifiable information. Patients’ online ID was used to limit duplicated reports. The questionnaire contained 22 questions, divided into the following three sections: the quality of life (QOL); the shift in symptoms and treatments during the COVID-19 pandemic; and Patient Health Questionnaire-9 (PHQ-9) (Appendix A). 

Each question regarding the QOL measures was derived based on a comprehensive discussion amongst subject matter experts, including surgeons, psychiatrists, nurses, researchers, and statisticians. These questions also referenced previously published studies and were modified to reflect the culture of South Korea. Patients answered each question on a 5-point Likert scale, ranging from 0–4, with 4 reflecting the poorest QOL; a score of 3 or 4 was categorized into poor QOL. Patients reported their residential areas in demographic variables, and they were later divided into two categories (urban or metropolitan area and suburban or rural area). 

In the PHQ-9 score analysis, we compared the PHQ-9 score of hyperhidrosis patients with that of the general population survey, which was performed during the same period under the supervision of the government. Korean people aged between 19 to 70 were randomly contacted through telephone call or online survey platforms. The selection of participants accounted for the population ratio per city among 17 metropolitan areas. The final cohort included 2063 participants (sampling error = ±2.2% at 95% confidence level). The general population survey expressed the PHQ-9 score [19] as a mean without the description of normality test. Thus, in this study, we attempted to express the results of hyperhidrosis patients as the mean and standard deviation for comparison. Due to the limited availability of participants data in this national report, a direct comparison with these values should be interpreted cautiously.

### 2.3. Statistical Analysis 

Continuous variables (age, body-mass index, the duration of hyperhidrosis, PHQ-9 score, time, hospital duration period) were described as the median and interquartile ranges (IQR), or the mean and standard deviation. Categorical variables were indicated as characteristics or percentages of patients based on the group. As we analyzed the continuous variables, the Mann–Whitney test or two-sample *t*-test was used after testing for normality via the Shapiro–Wilk method. The Chi-square test or Fisher’s exact test was used to analyze categorical variables. Due to the non-normality of data, Spearman correlation analyses were performed to investigate the correlation between PHQ-9 and age, BMI, or duration of the disease. Multiple linear regression was used to identify predictive factors related to the PHQ-9 score. We used R package version 4.0.5 for statistical analysis. *p*-values of less than 0.05 were defined as significant.

### 2.4. Ethical Statements

This study was approved by the Institutional Review Board of our institution (Approval No. 3-2021-0312). Participants’ consents were received through the online process, and their private information was protected under the supervision of the department.

## 3. Results

### 3.1. Characteristics of Participants 

A total of 222 patients participated in the questionnaire, and their PHQ-9 score was compared with that of the general population, which was based on the study by the Korean government (19). Half of the participants were female, and the median age of participants was 39.0 years (IQR 31.3–45.0). Two-thirds of them were from urban or metropolitan areas, and about half were married. The median duration of hyperhidrosis was 29.0 years; and more than two-thirds of them had palmar (72.5%) or plantar (73.9%) hyperhidrosis (Table 1).

### 3.2. PHQ-9 Score and Relationship with Hyperhidrosis 

The PHQ-9 score and the proportion of mild to moderate depression were demonstrated in Table 1 and Figure 1. The median PHQ-9 score was significantly higher in females (*p* < 0.001); the percentage of mild (50.5% vs. 30.6%, *p* = 0.004) and moderate (28.8% vs. 14.4%, *p* = 0.014) depression was also higher in females. Compared to the general population, male patients aged between 30–59 demonstrated a relatively lower mean PHQ-9 score (Figure 1). However, the mean PHQ-9 score in female patients was higher than the public, especially in the younger age groups (age between 20 to 39 years) (Figure 1). Furthermore, similar results (Figure 2) were observed after matching for patients’ age, gender, and disease duration (Appendix A).

The Spearman correlation showed a significant correlation between gender (ρ = +0.278, *p* < 0.001), age (ρ = −0.264, *p* < 0.001) or the duration of hyperhidrosis (ρ = −0.212, *p* = 0.001), and the PHQ-9 score. To understand what factors may influence the severity of depression, a multiple linear regression analysis was performed to identify possible predictors of the PHQ-9 score. Table 2 summarizes the results of the linear regression. The regression analysis indicated that female (β = +2.36, *p* = 0.006) residents in urban or metropolitan areas (β = −1.98, *p* = 0.017) and facial hyperhidrosis (β = +1.87, *p* = 0.024) are predictors for the PHQ-9 score.

### 3.3. The Effect of COVID-19 Pandemic on Patients’ Experience

#### 3.3.1. Quality of Life Measurements

With respect to the parameters related to the QOL (Figure 3), about half of all participants reported a poor rating in intimate personal contacts (60.4%), public speaking or presentation (58.6%), dressing (47.3%), the use of public spaces (45.9%), interpersonal relationship (44.1), and academic activity (43.2%) (Table 3). After matching for age, gender, and disease duration (Appendix A), females showed lower QOL in the use of public spaces (*p* = 0.030), public speaking or presentation (*p* = 0.027), and interpersonal relationship (*p* = 0.044) compared with male patients.

#### 3.3.2. Treatments and Symptoms

Most patients reported that symptoms stayed the same during the pandemic; only 10.4% of patients reported worsening and 4.1% experienced improvement (Figure 4). Additionally, about 70% of patients did not experience difficulty in getting treatment when compared with the pre-pandemic period. However, 11.7% of them complained of difficulty visiting the outpatient clinics due to restrictive measures or fear of contracting COVID-19, and 14% reported problems in getting appropriate medical information related to the disease. When patients were asked about their treatments during the pandemic, about half of them reported not receiving any treatments. Moreover, less than 20% of patients (outpatient visits, 15.4%; surgery, 1.4%) were managed under the guidance of specialists (Figure 5).

## 4. Discussion

The burden of mental problems after the COVID-19 pandemic has impacted the worldwide community, particularly those with predispositions. However, mental problems in patients with dermatologic conditions, such as hyperhidrosis, have not gotten much attention from physicians due to its relatively low severity. This study first tried to evaluate the impact of the COVID-19 pandemic on disease severity, treatment, and quality of life. Additionally, this study compared the level of depression in hyperhidrosis patients with that in the general population. 

The role of gender has been well recognized as a predictive factor for elevated depressive symptoms during the pandemic [20,21]. Compared to the general female population, female hyperhidrosis patients demonstrated a much higher PHQ-9 score. Moreover, male hyperhidrosis patients seemed to have a lower PHQ-9 score than female hyperhidrosis patients. It could be explained by relatively lower proportions of patients reporting poor QOL after matching for age and disease duration (Appendix A). Male patients had a significantly lower percentage of QOL-specific problems: the use of public spaces, interpersonal relationships, and public speaking or presentation. This could be attributable to different responses between male and female patients against restrictive social measures that minimized personal contacts. However, since there is not enough data on the pre-pandemic period, more consistent and serial studies would be warranted.

The location of hyperhidrosis is essential in predicting the quality of life and the improvement after treatments. Those with facial hyperhidrosis seemed to have reduced quality of life compared to those with palmar or axillary hyperhidrosis [22]; this could be related to the increased incidence of social phobia [23]. Even the benefit of endoscopic thoracic sympathectomy was not as rewarding in craniofacial hyperhidrosis as it was in other locations [24,25]. In this study, facial hyperhidrosis was found to be the risk factor for depressive symptoms. Due to these complex problems in craniofacial hyperhidrosis, much attention should be paid to the management of these patients, both in their sweating and mental problems.

Age and PHQ-9 score exhibited a negative relationship. A similar pattern was observed in the national analysis in Japan [26] among the general population; after middle adulthood, the PHQ-9 score decreased, though it increased in early adulthood. Additionally, hyperhidrosis patients with a longer disease duration seemed to have a low PHQ-9 score. It appears that patients may adjust their lifestyles and change their perception of the disease over time, as a similar psychosocial adjustment was observed in skin disease patients [27]. Further studies investigating the impact of disease period and age in hyperhidrosis would be necessary to find relevant factors to the process of psychosocial adaptation.

In terms of treatments during the pandemic, most hyperhidrosis patients did not report much difficulty in seeking treatments during the COVID-19 pandemic. This contradicts other medical or dermatological conditions that were impacted significantly due to measures to prevent the spread of SARS-CoV-2. Due to restrictions or impacts of the pandemic, patients experienced functional impairment in their quality of life [28]; marked reduction of face-to-face consultations, especially in dermatology [29]; delayed or under-reported diagnosis of skin malignancy [30]; and even decrease in urgent visit for dermatologic conditions [31]. However, this study showed that these kinds of impacts were not serious among hyperhidrosis patients; even the reduction in the outpatient visit might not be recognizable. It could be related to the negligence of hyperhidrosis among the medical professionals, as there is a limited number of good quality studies on this issue [32]. As this study showed, less than 20% of hyperhidrosis patients evaluated were under the guidance of medical professionals. Patients also complained of problems related to reliable information for the treatment and the lack of guidance. Many patients managed the disease by self-medication or non-medically-approved strategies. This lack of supervision could induce more complications, adverse events, or even life-threatening situations [33]. Thus, further involvement from professional medical staff would be warranted to improve the quality of care.

There are several limitations to this study. First, this was a cross-sectional study based on an online surveillance. Thus, it could not measure how the score of PHQ-9 has changed over time, and there is no data about their mental well-being before the pandemic. Therefore, further follow-up studies of these patients are warranted to better understand the impact of COVID-19 pandemic on the population suffering from hyperhidrosis. Second, a direct comparison between the study population with the general population [19] should be interpreted cautiously. Due to the lack of raw data, we could not compare the distribution of the PHQ-9 score, and it was statistically challenging to demonstrate the difference among hyperhidrosis patients. Third, due to the online questionnaire-based study design, there may be respondent bias. As we enrolled patients through the online community, patients interested in hyperhidrosis or those with a longer disease duration became more interested, and thus, participated in the study. To demonstrate more generalized results, a large-scale multinational study is warranted. Moreover, the questions used to assess the QOL of hyperhidrosis patients are yet under the process of validation and the QOL of the general population was not assessed as we performed. Therefore, the direct comparison of the QOL between hyperhidrosis patients and the public would be difficult and further studies are necessary.

Hyperhidrosis patients had a relatively higher level of depression during the pandemic, especially among female patients, and a proportion of hyperhidrosis patients complained of difficulty in treatment during the pandemic. Psychiatric interventions are warranted for female or facial hyperhidrosis patients. Further multi-institutional prospective studies should be done to better investigate the burden of mental distress among hyperhidrosis patients.

## Figures and Tables

**Figure 1 jcm-11-03576-f001:**
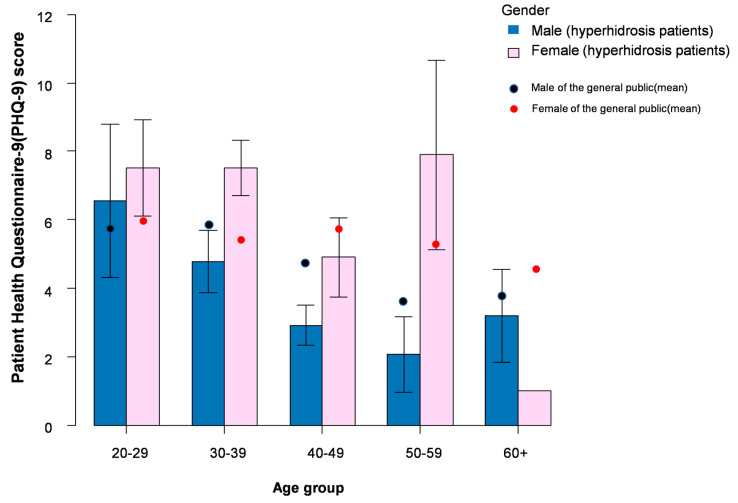
Patient Health Questionnaire –9 (PHQ –9) score for hyperhidrosis patients and the general population.

**Figure 2 jcm-11-03576-f002:**
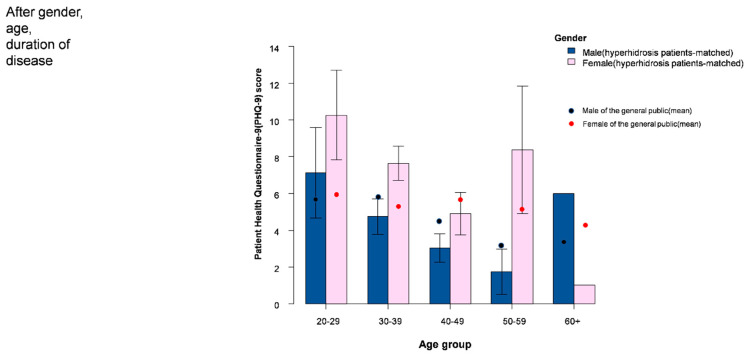
Adjusted Patient Health Questionnaire–9 (PHQ–9) score for hyperhidrosis patients and the general population (age, gender, and disease duration were adjusted).

**Figure 3 jcm-11-03576-f003:**
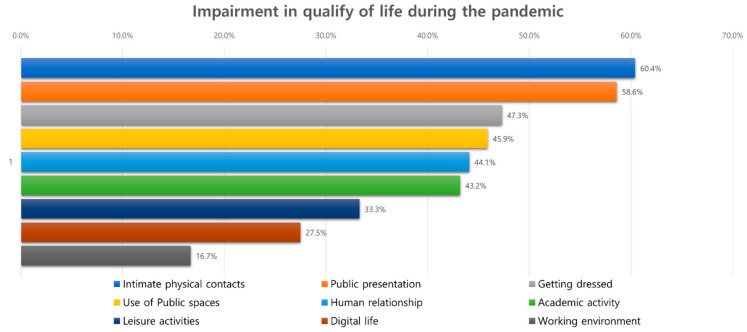
The proportion of hyperhidrosis patients with impairment in the quality of life.

**Figure 4 jcm-11-03576-f004:**
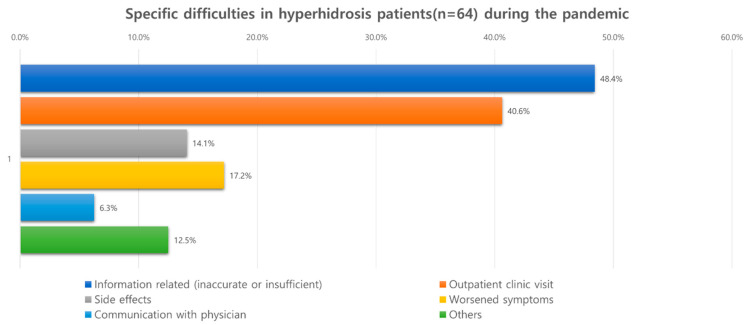
Problems that hyperhidrosis patients faced during the pandemic.

**Figure 5 jcm-11-03576-f005:**
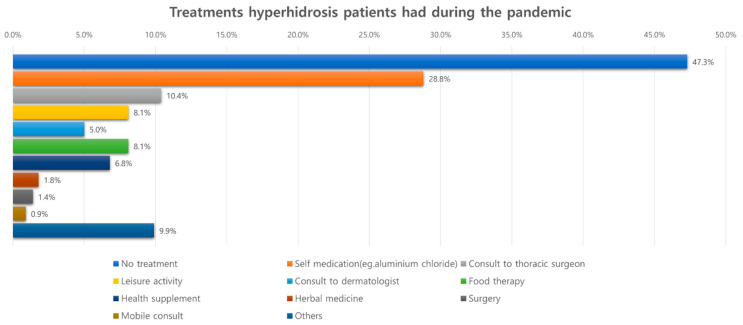
Treatments that hyperhidrosis patients experienced during the pandemic.

**Table 1 jcm-11-03576-t001:** Participants’ demographic, hyperhidrosis characteristics and PHQ-9 score.

	Total	Male	Female	*p* Value
	*n* = 222	*n* = 111	*n* = 111	
Demographics				
Age	39.00 [31.25, 45.00]	42.00 [34.00, 47.00]	37.00 [29.00, 43.00]	0.001
Age group				0.001
20–29	38 (17.1)	9 (8.1)	29 (26.1)	
30–39	77 (34.7)	37 (33.3)	40 (36.0)	
40–49	77 (34.7)	46 (41.4)	31 (27.9)	
50–59	24 (10.8)	14 (12.6)	10 (9.0)	
60+	6 (2.7)	5 (4.5)	1 (0.9)	
BMI	23.10 [21.20, 25.60]	24.70 [23.00, 26.70]	21.80 [20.15, 23.50]	<0.001
Marital status				0.033
married	109 (49.1)	63 (56.8)	45 (40.5)	
divorced	5 (2.3)	1 (0.9)	4 (3.6)	
unmarried	108 (48.6)	47 (42.3)	61 (55.0)	
Residential area				
Small town	72 (32.4)	38 (34.2)	34 (30.6)	0.667
Urban/Metropolitan area	150 (67.6)	73 (65.8)	77 (69.4)	
Depression related				
PHQ-9 mean (SD)	5.25 (5.97)	3.74 (4.89)	6.77 (6.56)	<0.001
PHQ-9 median (IQR)	3.5 [1.0, 8.0]	2.00 [0.00, 6.00]	5.00 [2.00, 10.00]	<0.001
Mild Depression ^¶^	90 (40.5)	34 (30.6)	56 (50.5)	0.004
Moderate Depression ^⁑^	48 (21.6)	16 (14.4)	32 (28.8)	0.014
Hyperhidrosis related				
Duration of disease	29.0 [20.0, 34.0]	30.00 [20.00, 35.00]	26.00 [20.00, 33.50]	0.061
Location of hyperhidrosis				
Axilla	111 (50.0)	45 (40.5)	66 (59.5)	0.007
Palmar	161 (72.5)	76 (68.5)	85 (76.6)	0.229
Plantar	164 (73.9)	78 (70.3)	86 (77.5)	0.285
Facial	82 (36.9)	44 (39.6)	38 (34.2)	0.487
Multiple	188 (84.7)	88 (79.3)	100 (90.1)	0.039
Other areas	42 (18.9)	20 (18.0)	22 (19.8)	0.864
Comorbidity ^⁂^				
Gastrointestinal diseases	84 (37.8)	36 (32.4)	48 (43.2)	0.128
Obesity	62 (27.9)	33 (29.7)	29 (26.1)	0.654
Anxiety	44 (19.8)	18 (16.2)	26 (23.4)	0.238
Depression	44 (19.8)	15 (13.5)	29 (26.1)	0.028
Hypertension	27 (12.2)	22 (19.8)	5 (4.5)	0.001
Diabetes mellitus	8 (3.6)	6 (5.4)	2 (1.8)	0.280
Neurologic disorder	17 (7.7)	9 (8.1)	8 (7.2)	1

BMI, body-mass index; SD, standard deviation; IQR, interquartile range; PHQ-9, patient health questionnaire-9. ^¶^ Patients with PHQ-9 score 5 or above. ^⁑^ Patients with PHQ-9 score 10 or above. ^⁂^ This is based on the patients’ self-report not previous medical records.

**Table 2 jcm-11-03576-t002:** Predictors for PHQ-9 score, identified by multiple linear regression analysis.

Factor	Estimate	Standard Error	t Value	*p* Value
(Intercept)	8.16 (0.39 to 15.93)	3.94	2.07	0.040
Age	−0.08 (−0.18 to 0.02)	0.05	−1.67	0.096
BMI	−0.06 (−0.31 to 0.19)	0.13	−0.46	0.640
Marital status [single or divorced]	1.66 (−0.16 to 3.48)	0.92	1.8	0.074
Gender [female]	2.36 (0.7 to 4.03)	0.84	2.79	0.006
Residential area [urban/metropolitan]	−1.98 (−3.61 to −0.35)	0.83	−2.4	0.017
Axillary hyperhidrosis	0.63 (−0.9 to 2.17)	0.78	0.81	0.420
Facial hyperhidrosis	1.87 (0.25 to 3.5)	0.83	2.27	0.024

BMI, body-mass index; PHQ-9, patient health questionnaire-9.

**Table 3 jcm-11-03576-t003:** Quality of life and patients’ experience during the pandemic.

	Total	Male	Female	*p* Value
	*n* = 222	*n* = 111	*n* = 111	
Quality of life problems				
Use of public spaces, (%)	102 (45.9)	43 (38.7)	59 (53.2)	0.043
Working environments (%)	37 (16.7)	12 (10.8)	25 (22.5)	0.030
Interpersonal relationship (%)	98 (44.1)	38 (34.2)	60 (54.1)	0.004
Public speaking or presentation (%)	130 (58.6)	52 (46.8)	78 (70.3)	0.001
Digital life ^¶^ (%)	61 (27.5)	25 (22.5)	36 (32.4)	0.132
Intimate personal contacts (%)	134 (60.4)	59 (53.2)	75 (67.6)	0.039
Dressing (%)	105 (47.3)	43 (38.7)	62 (55.9)	0.015
Hobbies and leisure activity (%)	74 (33.3)	30 (27.0)	44 (39.6)	0.064
Academic activity (%)	96 (43.2)	40 (36.0)	56 (50.5)	0.042
Change in symptoms				0.750
Improved	9 (4.1)	5 (4.5)	4 (3.6)	
Same as usual	190 (85.6)	96 (86.5)	94 (84.7)	
Worsened	23 (10.4)	10 (9.0)	13 (11.7)	
Problems in treatments during the pandemic				
No problem	158 (71.2)	82 (73.9)	76 (68.5)	0.459
Information related issues	31 (14.0)	15 (13.5)	16 (14.4)	1
Inaccurate information	12 (5.4)	8 (7.2)	4 (3.6)	0.374
Insufficient information	26 (11.7)	11 (9.9)	15 (13.5)	0.532
Visit to outpatient clinics	26 (11.7)	13 (11.7)	13 (11.7)	1
Side effects	9 (4.1)	4 (3.6)	5 (4.5)	1
Communication with physicians	4 (1.8)	3 (2.7)	1 (0.9)	0.622
Treatments received during the pandemic				
No treatment	105 (47.3)	62 (55.9)	43 (38.7)	0.015
Self-medication ^⁂^	64 (28.8)	23 (20.7)	41 (36.9)	0.011
OPD visit to thoracic surgeons	23 (10.4)	10 (9.0)	13 (11.7)	0.660
OPD visit to dermatologist	11 (5.0)	5 (4.5)	6 (5.4)	1
Surgery	3 (1.4)	3 (2.7)	0 (0.0)	0.247
Diet therapy	18 (8.1)	8 (7.2)	10 (9.0)	0.807
Leisure activity	18 (8.1)	7 (6.3)	11 (9.9)	0.462
Herbal medicine	4 (1.8)	1 (0.9)	3 (2.7)	0.622
Mobile consultation	2 (0.9)	0 (0.0)	2 (1.8)	0.498
Health supplement	15 (6.8)	6 (5.4)	9 (8.1)	0.594
Others	22 (9.9)	9 (8.1)	13 (11.7)	0.501

OPD, outpatient department. ^¶^ It includes inconvenience related to the use of smartphone and electronic devices. ^⁂^ Over-the-counter drugs, aesthetic products, and the use of topical agents such as aluminum chloride.

## Data Availability

The data that support the findings of this study are available from the corresponding author upon reasonable request.

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
