# Peer review of "The Impact of COVID-19 on Hyperhidrosis Patients in the Mental Health and Quality of Life: A Web-Based Surveillance Study"

_jcm, 2022, doi:10.3390/jcm11133576_

Round 1

Reviewer 1 Report

Its an interesting study which highlights the quality of life and the degree of depression among different groups of HH patients and the influence of  the pandemic when patients could not access medical care.

I found this study worthwhile:

-as the HH population in the study is large,

-the mental health questionnaire results comparing different subgroup of HH ( facial, axillae and palmar plantar) are interesting and original.

-The fact that the facial HH group showed more vulnerability to depression has never been published as far as I know. It showed that the importance of treating this group of patients actively.

-More is published on palmar plantar and axillae HH as they are more frequent than on facial HH.

-The methodology is good and analysis of results satisfactory

Reviewer 2 Report

This is an interesting cross-sectional study regarding the psychological impact of the COVID-19 outbreak in hyperhidrosis patients. The paper could be improved with some minor changes:

-In the online survey subsection of the methods section it would be interesting to add more information about the general population PHQ-9 survey, such as its development and recruitment of subjects. 

-Did the authors evaluate the study level of the subjects? Usually in association-based online survey studies the included subjects have a higher level regarding the general population.

-Were the QOL questions validated? If not, please state as a limitation. Please also report that these questions were not compared with the general population group.
